# How to improve representativeness and cost-effectiveness in samples recruited through meta: A comparison of advertisement tools

**Anja Neundorf**, **Aykut Öztürk***

School of Social and Political Sciences, University of Glasgow, Glasgow, United Kingdom

* aykut.ozturk@glasgow.ac.uk

## Abstract

The use of paid advertisements on social media, in particular Meta platforms, to create samples for online survey research is becoming increasingly common. In addition to researchers working on hard-to-reach populations, Meta's promise of unmediated, quick, and cheap access to a large pool of survey takers across the world is appealing also for researchers who want to create diverse samples of national populations for cheaper prices. Yet the design of Meta's advertisement optimization algorithm complicates the use of Meta advertisements for this purpose, as it generates a trade-off between cost-effectiveness and sample representativeness. In this paper, we rely on original online surveys conducted in the United Kingdom, Turkey, Spain, and the Czech Republic to explore how two primary tools determining the audience of Meta advertisements, i.e., campaign objectives and demographic targeting, affect the recruitment process, response quality, and sample characteristics. In addition to documenting the trade-offs between the cost and representativeness in Meta samples, our paper also shows that researchers can create high-quality, cost-efficient, and diverse samples if they use the right combination of Meta advertisement tools.

## Introduction: Potentials and pitfalls in recruitment through meta advertisements

The use of social media as a tool for the recruitment of survey participants is becoming increasingly common among social scientists. In addition to its uses for recruiting participants from niche and traditionally hard-to-reach populations [1–3], paid advertisements running on social-media platforms also appeal to scholars whose goal is to create diverse samples of the general population for the study of public opinion, especially through experimental methods. During the last few years, many scholars have used Meta Ads for this purpose in countries all around the world, such as Egypt [4], Indonesia [5], Kenya and Tanzania [6], Turkey [7], and Uruguay [8], studying topics ranging from populist and xenophobic attitudes to preferences for democratic or authoritarian regimes.

This interest in Meta paid advertisements is not surprising, given the reach of the platform. Meta facilitates advertising on the most popular social-media platforms in the world, including

**Data Availability Statement:** Data relevant to this work are available from the Harvard Dataverse at DOI:10.7910/DVN/YEOX5N (https://doi.org/10.7910/DVN/YEOX5N).

**Funding:** AN Grant number: 865305 European Research council The funders had no role in study design, data collection and analysis, decision to publish, or preparation of the manuscript.

**Competing interests:** The authors have declared that no competing interests exist.

Facebook, Instagram, and Facebook Messenger, simultaneously. Fig 1 presents the percentage of each country's population that is active just on Facebook, the leading social media platform in the world, at least once per month. The data are from 2019 and totalled 2.5 billion, one-third of the world population [9]. As the figure confirms, in both democratic and authoritarian countries Facebook reaches a large proportion of the population.

Despite this impressive size and diversity of Meta's user base, however, samples recruited through Meta might end up being highly skewed. Researchers using Meta advertisements report significant biases sometimes towards men [e.g., 10] and sometimes towards women [e.g., 11], sometimes towards older people [e.g., 12] and sometimes towards younger people [e.g., 13], and always towards highly educated people [14]. These biases lead some scholars to question the value of using Meta advertisements for recruitment purposes [15].

Bias in Meta samples is partly a self-selection problem, which haunts all online opt-in survey designs. Some demographic groups are more likely to use Facebook on a regular basis, and some demographic groups are more interested in and comfortable with sharing their opinions in online surveys. However, an important part of the problem is unique to the way Meta advertisements work. Rather than assigning its advertisement space randomly, Meta uses powerful optimization algorithms that deliver advertisements to the Meta users who will be most likely to take certain actions, such as clicking on the advertisement [16]. These algorithms serve business owners well, but they can significantly exacerbate the selection bias for researchers. Furthermore, given that the cost of Facebook samples is dependent on choices made by advertisers, trying to avoid these algorithms altogether might result in samples that are more expensive than other data-collection methods, defying the purpose altogether. There are instances in which data collection through Meta platforms has been reported to be more expensive than other forms of recruitment [e.g., 17, 18].

We believe that researchers can leverage the power of Meta's advertisement algorithms to create more representative samples if they develop a better understanding of the tools offered by Meta. In this paper, we study two principal tools that determine the audience of Meta advertisement campaigns: campaign objectives and demographic targeting. The former tool allows advertisers to determine which optimization algorithm Meta will use, while the latter allows advertisers to limit an advertisement to a pre-defined demographic group. Our study explores

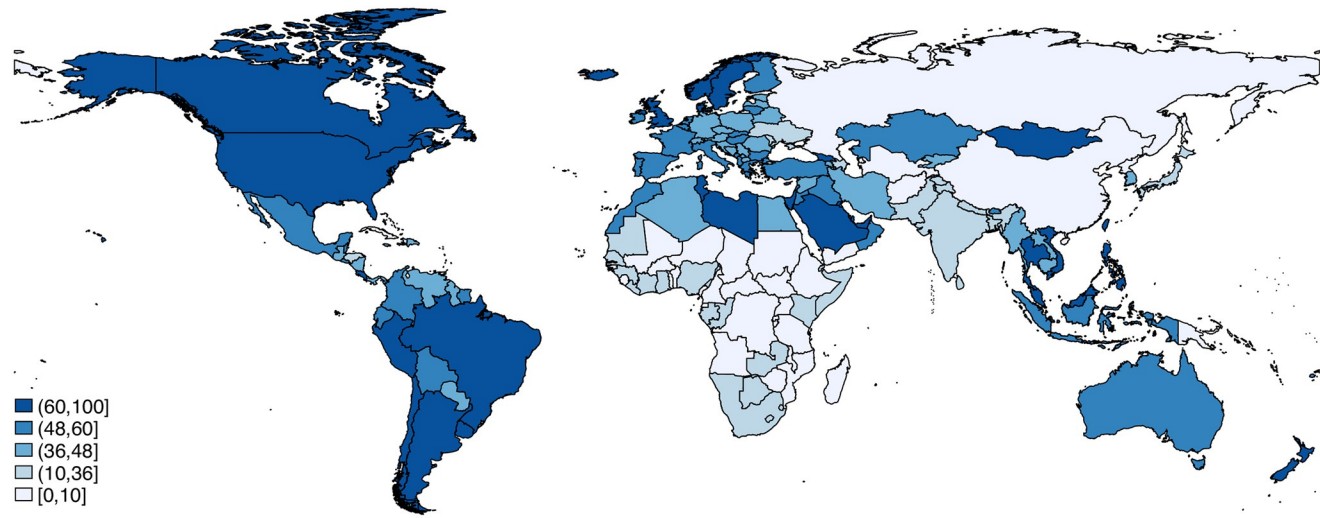

**Fig 1. Percentage of population that is active on Facebook at least once per month (2019).** Source for the data: World Population Review (2021).

how various combinations of these two tools affect sample characteristics, response quality, and overall cost, by relying on Meta advertisement campaigns and original online surveys conducted in the United Kingdom (UK), Turkey, Spain, and the Czech Republic.

Our findings demonstrate that scholars can use campaign objectives that track Meta users outside of the Meta environment to deliver their advertisements more effectively. The conversion objective, which is about to be renamed by Meta as sales and engagement objectives, helps researchers to separate "survey completers" from "link clickers", resulting in significantly lower costs of participant recruitment. It is important to underline that these gains do not come at the expense of sample quality. We do not find substantial differences between conversion samples and other samples with respect to response quality and sample biases: conversion campaigns recruit more people than other campaigns even in harder-to-reach groups. Reach campaigns, on the other hand, emerge as completely ineffective at recruiting survey respondents.

Second, our research documents how the use of demographic targeting affects sample sizes and sample characteristics. We show that targeting by a single demographic can produce a more diverse sample than not using any targeting, and, in our experience, the former is usually not even more expensive than the latter. More specific targeting, such as advertisements separately targeting intersections of gender, age, and education subgroups, increases costs but also substantially improves sample representativeness. Thus, combining optimization algorithms targeted towards survey completers with some levels of demographic targeting emerges as the best recruitment strategy for researchers using Meta advertisements.

The rest of this paper is formed of four sections. First, we introduce the Meta tools that we study in this paper. In the second section, we describe our study materials and methods. The third section presents our findings. We conclude by discussing our findings, limitations, and the impending changes to online advertisement.

## Meta advertisement tools

Meta offers a range of tools to shape the delivery and the audience of advertisements. We study two of the most powerful Meta advertisement tools: the selection of campaign objectives and the use of targeting.

### Campaign objectives

Each advertiser has to choose a campaign objective to create advertisements on Meta. Meta defines campaign objectives as, "what the advertiser wants people to do when they see the advertisement" [19]. The choice of campaign objectives determines which optimization algorithms will then be available to reach these objectives. Meta offers a set of campaign objectives which fall under three main categories: reach, traffic, and conversion.

Reach campaigns aim to deliver advertisements to the highest number of Meta users [19]. These campaigns are especially recommended for brand recognition purposes. Traffic campaigns aim to increase the number of link clicks on advertisements. After a phase of learning, during which Meta delivers the advertisement broadly to learn who will show interest in it, the Meta algorithm focuses its delivery on user groups that are more likely to click on the advertisement.

Conversion campaigns aim to increase the number of user actions taken outside of Meta platforms, such as online purchases from the advertiser's website. These campaigns track Meta users' behavior on websites defined by the advertiser, and then send information to Meta to improve the advertisement delivery. Thus, for example, a researcher can tell Meta to show her advertisements to users who will be more likely to complete her survey hosted on the Qualtrics

survey platform. In this case Meta will first show her advertisements broadly on Meta's social-media platforms, including Facebook and Instagram, and track which users are completing the survey on Qualtrics. After the learning phase, Meta will start delivering advertisements to its users who will be more likely to complete the survey.

According to our review of the literature, "traffic" has been the most popular choice of campaign objectives for researchers. This decision is sometimes justified with reference to previous works [20]. Conversion campaigns are relatively new: they were introduced only in 2017. Also, it is more complicated for a new user to use the conversion objective compared with using other campaign objectives. While a user can launch her first traffic campaign in minutes, it can take several hours to install the required tools to make the conversion objective work. For a more detailed explanation of how to set up advertisement campaigns on Meta, please see [21].

### Targeting

Meta allows advertisers to limit the target audience of their advertisement sets to certain categories predefined by Meta [21]. These categories range from demographic characteristics, such as age and gender, to hobbies and interests. Researchers can use this tool to create different versions of the same advertisement, each with a distinct target audience, to ensure that the advertisement campaign recruits participants from all of the targeted demographic categories [14, 22]. However, the trade-off between cost and diversity is again at play here. Dividing the advertisement campaign budget into more advertisement sets slows down Meta's machine learning algorithms. Slower learning phase for Meta will result in less efficient advertisements and increased costs. Meta openly warns advertisers that this use of targeting will negatively affect advertisement campaigns' performances [23].

Researchers have, of course, different goals from advertisers, and they can tolerate a certain amount of increase in cost depending on their research goals and alternatives. However, it is important to document these trade-offs, as the cost is still an important consideration: a very expensive advertisement campaign might defy the purpose of using Meta for recruitment purposes.

## Materials and methods

### Data

Our study was conducted in four countries, i.e., the UK, Turkey, Spain, and the Czech Republic, as part of a larger research survey project. Our online survey consisted of 26 questions and included a randomized survey experiment. All of the surveys were conducted in the official and the most spoken language of the respective country, i.e., English, Turkish, Spanish, and Czech, allowing us to maximize the size of our audience. The design of our study was pre-registered on the Open Science Foundation platform on February 8, 2021, before any data collection commenced. More information about the pre-analysis plan and the ethical approval can be found in Section 1 of S1 File. Our first survey was conducted in the UK between February 7 and February 15, 2021, and the data collection for the last survey ended on April 7, 2021.

Our cases represent a diverse selection in terms of political, economic, and demographic properties. The UK constitutes a typical case of an advanced democracy, and it is the second most-studied country in political science [9, 24]. However, many researchers are increasingly using Meta advertisements to study less developed, non-democratic countries, where data collection is more restricted. Turkey represents such a context. It is a middle-income country that was, according to the influential Varieties of Democracy dataset, an electoral authoritarian regime by the time our study was conducted. Spain and the Czech Republic are younger democracies that are located between the UK and Turkey in respect of their levels of gross

domestic product (GDP) per capita. The Czech Republic is also distinguished by its relatively small population, which amounts to only 10 million people. While there are only around 5 million Facebook users in the Czech Republic, Facebook users amount to 30 million people in Spain (61% of the population) and 44 million in Turkey and the UK (respectively 53% and 64% of the national populations).

We compared six different advertisement campaigns in each of these countries. In terms of the sample size, our target was to continue recruitment for all campaigns until we reached at least 250 respondents through each advertisement campaign. This would allow us to investigate the demographic characteristics of each sample. We decided to stop data collection for all campaigns once we reach 1,800 successful survey completions in total, which is 20% larger than the targeted sample size for this study, even if we failed to recruit 250 participants through some campaigns. In line with this expectation, more than one campaign in all four countries failed to recruit the targeted 250 participants, forcing us to continue the data collection until we reached 1,800 successful participants per country in total. Importantly, this design ensured that all campaigns running in the same countries stayed open for the same amount of time and spent the exactly same amount of budget, making it comparable with other campaigns in the same country. On the other hand, the duration of campaigns running in different countries may vary; for example, campaigns in Spain stayed open longer than campaigns in Turkey as it took more time to recruit enough respondents in the former.

## Study design

As a whole, our study compares the three campaign objectives introduced in the previous section: reach, traffic, and conversion. As we discuss in more detail below, campaigns using the reach campaign objective failed to recruit enough participants in the UK. After evaluating this finding, which was consistent with the results from the pilot studies, and finding it to be conclusive, we dropped the reach campaign objective from our studies and instead focused more on variations of targeting strategies.

We tested the use of two types of targeting strategies, in addition to the use of no-targeting. In our first survey, conducted in the UK, we compared a "no-targeting" strategy to a "cross-targeting" strategy. Under the cross-targeting strategy, we crossed subgroups of three age groups (young, middle-aged, old), two gender categories (female, male) and three education categories (college graduates, high school graduates, and people with unspecified education status on Meta). In total, we created 18 different advertisements, each with a £2 daily spending budget. Under the no-targeting strategy, on the other hand, we assigned the entire daily campaign budget, i.e., £36, to one single advertisement without defining any demographic targets for that advertisement.

In our subsequent studies in Turkey, Spain, and the Czech Republic, we added a new demographic targeting strategy, which we call "single targeting." In this method of targeting, we targeted each of the eight groups listed above (young, middle-aged, old, female, male, college graduates, high school graduates, and people with unspecified education status on Meta) separately, without crossing their subgroups. In total, we had eight different advertisements, each starting with a daily budget of £4.50. Table 1 lists the eight different combinations of campaigns objectives and targeting that are compared in our study.

We used the same advertisement image and text for all the advertisement campaigns, which was designed as a generic advertisement similar to the ones frequently used by previous researchers (see [21] for images of our advertisement and [25] for the effects of different images). No incentives were used for recruiting participants.

**Table 1. Combinations of targeting and campaign objectives explored in each of our cases.**

|            |        | Campaign Objectives | | |
|------------|--------|:----------:|:-------:|:-----:|
|            |        | **Conversion** | **Traffic** | **Reach** |
| **Targeting** | None | *U,T,S,C* | *U,T,S,C* | *U* |
|            | Single | *T,S,C* | *T,S,C* | *N/A* |
|            | Cross  | *U,T,S,C* | *U,T,S,C* | *U* |

U: United Kingdom; T: Turkey; S: Spain; C: Czech Republic

Once participants saw our advertisements on their social media feed (on Facebook, Facebook Messenger, or Instagram), they decided whether to click the advertisement link or not. If they clicked on the advertisement, they were directed to the landing page of the survey, which was programmed in Qualtrics. The first page these Meta users saw was the consent page, which informed them about the identity of the researchers, the purposes of our research, and how we would use participant data. Respondents could see survey questions only if they gave consent to participate in our study and they were at least 18 years old. Participants were then asked a series of demographic and political questions, which we analyze below. All participants successfully completing the survey were directed to a "Thank You" page hosted on a WordPress website. This website sent information to Meta Ads Manager about successful completions of conversion campaigns, allowing the Meta algorithm to recruit more completers [21]. Our analysis below uses data collected through the Meta Ads Manager (e.g., costs) as well as from the substantive survey that participants took.

## Outcome measures

We compared our samples, recruited through each of our advertisement campaigns, across three groups of outcome measures: overall cost, sample representativeness, and response quality.

**Overall cost.**   To compare the overall cost of surveys, we report the cost per *completed* survey across each sample. This equals the division of the total cost of a sample into the number of completed surveys in that sample. The total cost is the money we paid to Meta, excluding taxes, for the advertisement campaign recruiting that particular sample. The data on the number of completed surveys, on the other hand, come from the analysis of the sample downloaded from Qualtrics.

**Sample representativeness.**   In order to explore sample representativeness, we compare characteristics of each of our samples to benchmarks received from European Social Survey (ESS) and the Comparative Study of Election Studies (CSES). We use two groups of variables in these comparisons: variables that were targeted in our surveys by Meta tools and variables that were not targeted but which were theoretically important for our substantive research question. The targeted variables, as detailed in the section above, were age, gender, and education. By studying demographic changes in these categories, we explore to what extent, and at what level of cost, targeting improves sample representativeness along targeted variables.

It might also be the case that campaign objectives and targeting tools created biases in variables that were *not* directly targeted through Meta tools. Researchers should be especially careful if they have reason to expect that their mode of data collection will create biases towards a variable that is theoretically important [26]. We have two such variables in our own project: political interest and partisanship. People who are partisan or politically interested might be more willing to express their political opinions when given the chance, and online surveys give

people chances to do so [27]. Thus, we also explore whether certain campaign objectives aggravate these biases.

**Response quality.** Measuring response quality is especially important in the context of campaign objectives. It is possible that certain algorithms deliver advertisements to more or less attentive Meta users. We use four metrics to measure the data quality and attentiveness of respondents: passing an instructional attention check, survey duration, engagement level with an open-ended question, and providing contact information at the end of the survey.

Instructional attention checks are one of the most common ways used to measure whether respondents are paying attention to surveys [28]. Following this practice, we used statements that asked participants to choose a particular response option. Assuming that straightlining in particular would be an issue in our survey, which used several matrix questions, we located the attention check as the last statement in a three-item matrix question [29]. Furthermore, as we expected that inattentive behavior would especially be common toward the last part of the survey, we showed our attention check question only after respondents had completed 75% of the survey. A screenshot of the attention-check used in the UK is provided in Section 3 in S1 File.

Our second measure of response quality is survey duration. We generally expect that surveys completed in very short time periods mean low quality responses. Below, we compare our samples with respect to both mean and median value for survey duration.

Our third measure of quality refers to the length of open-ended answers [30, 31]. This measure helps us to see the cognitive and physical effort a respondent has put in while answering our questions. In line with the substantive focus of our survey, we asked respondents to write down what they believed "democracy" means and measured how many words the average respondent used to answer that question.

Finally, at the end of our survey, we asked our respondents if they were willing to share their WhatsApp number or e-mail address, so that re-contact would be possible. We believe that a respondent will feel motivated to take another survey only if they are taking the current survey seriously. Thus, this can be taken as an indirect measure of response quality. The proportion of participants providing follow-up information is also important for researchers who want to create panel data to measure long-term effects and changes.

## Results

We summarize our findings under two sections: first for campaign objectives and then for demographic targeting.

### How campaign objectives affect participant recruitment

**Overall cost.** Table 2 presents the results from conversion, traffic, and reach campaigns in all four countries. The analysis of campaign objectives reveals that Meta delivers what it promises with respect to campaign objectives: reach campaigns are shown to the most Meta users, traffic campaigns provide the most link clicks on the Meta advertisement, and conversion campaigns produce the highest amount of successful survey completion.

To begin with, as can be seen in Row A in Table 2, our advertisements using the reach campaign objective were by far the ones seen by the most Meta users in the UK. Eleven times more people saw the ad when "reach" rather than "conversion" was used as the campaign objective. However, only a very small percentage of these Meta users actually clicked on our links, and only six of them completed the survey. We believe this is because the Meta algorithm shows advertisements under the reach category only to *Meta scrollers*, i.e., Meta users who do not usually click on links. In our view the evidence is conclusive that the reach campaign objective

**Table 2. Comparison of campaign objectives.**

|   |   | Conversion | Traffic | Reach |
|---|---|---:|---:|---:|
|   | **United Kingdom (data collection: Feb 9–15, 2021)** |   |   |   |
| A | N of Meta users who saw the ad on Meta | 56,486 | 82,480 | 655,815 |
| B | N of Meta users who clicked on the ad | 1,597 | 1,598 | 194 |
| C | N of respondents who completed the survey | 1,032 | 756 | 6 |
|   | Click-through rate | 2.80% | 1.90% | 0.03% |
|   | Completion rate | 1.80% | 0.90% | 0.00% |
|   | Average cost per completed survey | £0.33 | £0.44 | £54.50 |
|   | **Turkey (data collection: Feb 19–22, 2021)** |   |   |   |
| A | N of Meta users who saw the ad on Meta | 498,561 | 1,030,721 |   |
| B | N of Meta users who clicked on the ad | 4,603 | 17,906 |   |
| C | N of respondents who completed the survey | 1,678 | 203 |   |
|   | Click-through rate | 0.92% | 1.74% |   |
|   | Completion rate | 0.34% | 0.02% |   |
|   | Average cost per completed survey | £0.21 | £1.76 |   |
|   | **Spain (data collection: Mar 1–15, 2021)** |   |   |   |
| A | N of Meta users who saw the ad on Meta | 314,494 | 985,910 |   |
| B | N of Meta users who clicked on the ad | 4,182 | 15,020 |   |
| C | N of respondents who completed the survey | 1,709 | 92 |   |
|   | Click-through rate | 1.32% | 1.52% |   |
|   | Completion rate | 0.54% | 0.01% |   |
|   | Average cost per completed survey | £0.87 | £16.15 |   |
|   | **The Czech Republic (data collection: Mar 24-Apr 7, 2021)** |   |   |   |
| A | N of Meta users who saw the ad on Meta | 247,745 | 579,769 |   |
| B | N of Meta users who clicked on the ad | 3,949 | 12,068 |   |
| C | N of respondents who completed the survey | 1,632 | 193 |   |
|   | Click-through rate | 1.59% | 2.08% |   |
|   | Completion rate | 0.66% | 0.03% |   |
|   | Average cost per completed survey | £0.89 | £7.51 |   |

*Note 1*: Completed surveys: United Kingdom = 1,794; Turkey = 1,892; Spain = 1,801; Czech Republic = 1,827.

*Note 2*: Click through rate: Proportion of link clicks to Meta users seeing the ad.

*Note 3*: Completion rate: Proportion of survey completions to Meta users seeing the ad.

does not help to recruit survey participants. We therefore did not use this campaign objective in the other countries.

The traffic campaign objective, as advertised by Meta, usually produced more link clicks on our advertisements than the other two campaign objectives. This can be seen by comparing the number of Meta users who clicked on the advertisements and the click-through rates (CTR). While conversion and traffic produced similar numbers of link clicks in the UK, in the remaining three countries traffic produced several times more link clicks with higher CTRs. Importantly, however, successful survey completers formed only a small proportion of respondents who saw our advertisements through traffic campaign objectives, as it can be seen through the completion rates. Meta users recruited through traffic campaigns were instead more likely to deny consent or to leave the survey incomplete.

The average costs were especially low in Turkey, where one completed survey cost around £0.21 (excluding sales taxes) when we used conversion as our campaign objective. In the UK, one completed survey could cost as low as £0.25 (using non-targeted conversion), which is

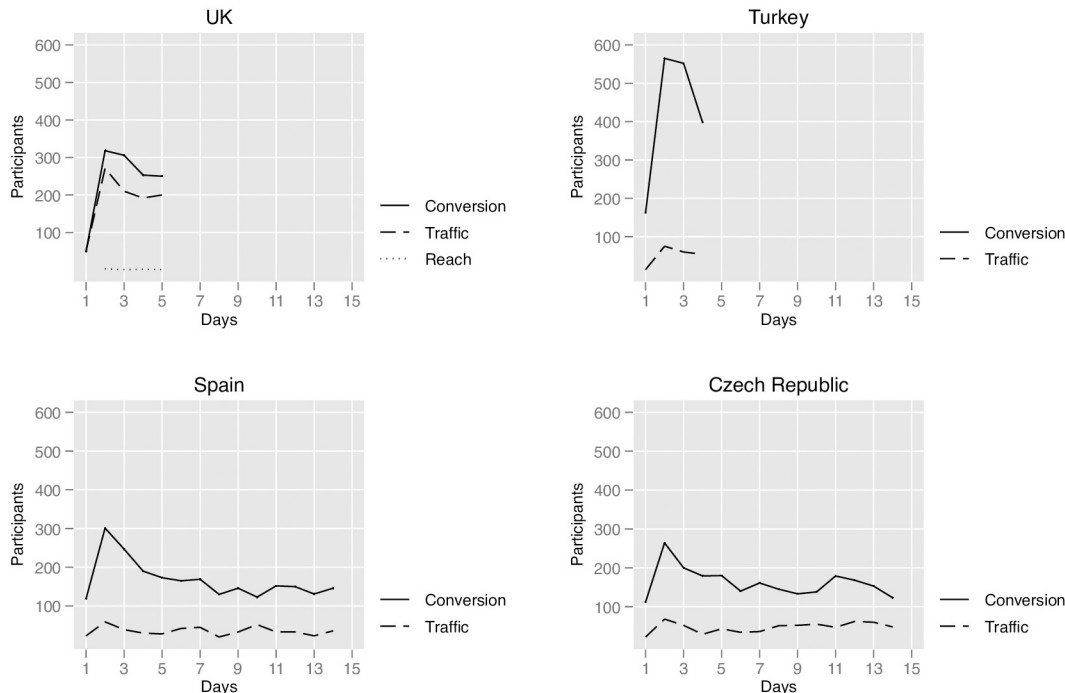

**Fig 2. Number of participants recruited per day.** *Note*: Completed surveys only. Data collection started at 5pm (GMT) on the first day of data collection; ads were closed at different times during the last day as the sample size reached 1,800 participants. The graph does not report the number of recruited participants on the last day, as this varies by the time the survey was closed.

about 18 times lower than a recent similar online survey conducted by one of the authors in the United States and UK with a commercial survey company, where the cost per respondent was £4.45. Even in Spain and the Czech Republic, where it cost significantly more to recruit participants compared to the UK and Turkey, a completed survey cost only £0.87 and £0.89 respectively when using conversion as the campaign objective.

The differences in average cost per country are driven by the number of days it takes to reach the target sample size. Meta will spend the daily budget as a default. The longer it is necessary to keep the ads open to recruit participants, the more expensive the overall data collection will be. Fig 2 plots the number of completed surveys per day for the different campaign objectives in our four cases. Section 2 in S1 File further provides the number of completed surveys per day for each of the targeting tools. As the figures illustrate, the recruitment process was fastest in Turkey and the UK, which have larger Meta user pools than Spain and especially the Czech Republic. This helped to reduce costs in the UK and Turkey. In all cases, conversion recruited more daily survey participants throughout the entire data collection.

Results regarding survey completion, cost, and speed of participant recruitment from our four countries unequivocally demonstrate the superiority of conversion to recruit survey participants. Although we spent the same amount of money for all the campaigns, conversion campaigns always ended up recruiting more successful survey completions. This was not because more Meta users saw or clicked on the advertisements from conversion campaigns, but because participants recruited through conversion campaigns were always more determined to complete the survey.

The gap between the conversion and traffic campaigns was clear even in our first survey in the UK. This difference further grew in the remaining countries to the extent that the traffic

campaign objective sometimes produced very few successful survey completions at very high cost, despite providing a significant amount of traffic to our Qualtrics page. We believe this is because Meta's tracking algorithm improved as we conducted surveys in different countries, successfully distinguishing "link clickers" from Meta users who would actually complete our surveys. In other words, Meta used the budget we assigned for traffic campaigns to recruit link clickers, while the budget for conversion campaigns was used to recruit survey completers.

**Sample representativeness.**    Are cheaper prices via conversion achieved at the expense of sample representativeness? Does conversion beat traffic and reach campaign objectives only because it channels all its resources to a few demographic groups that are better at completing tasks? In order to answer these questions, Table 3 compares the sample characteristics of the traffic and conversion campaign objectives with population characteristics taken from nationally representative surveys (the ESS and the CSES), based on the proportion of demographic groups that were harder to reach. We further conducted chi-square tests to compare the distributional properties of our samples with the population benchmark samples.

First of all, Table 3 reveals some patterns that were common for all the countries in our study. Our advertisement campaigns were less successful at recruiting younger, female, and non-college-educated people. Furthermore, Meta users completing our surveys had more interest in politics compared with the general population.

Importantly for our research question in this paper, the demographic and political differences between the participants in the conversion and traffic campaigns in our UK sample—the only sample in which traffic campaigns recruited a significant number of participants to allow for group comparisons and statistical analyses– seem to be quite small. Conversion samples fare even slightly better with respect to the age distribution, while traffic samples are slightly better at recruiting non—college graduates and females. Thus, preferring a conversion objective to a traffic objective does not lead to a significant demographic or political bias. Differences between conversion and traffic samples significantly grow in other countries, as can be seen in Table 3. Conversion is still better at recruiting younger people, but traffic samples are now less educated and less interested in politics, better reflecting population demographics. However, it is important to note that traffic samples in these countries are between eight and 18 times smaller than conversion samples, making the absolute numbers of less-educated and female respondents still smaller than those in the conversion samples. For example, conversion returns 523 non-college respondents in Turkey, while traffic returns only 119 respondents at the same cost.

**Response quality.**    The results of our quality checks are reported in Table 4. Pass rates in our surveys ranged from 79% in Turkey to 93% in the UK. These results are similar to results from commercial online panels in advanced countries. For example, using a similar attention-check question, [28] reports a 90% pass rate in an online survey conducted in the United States among respondents coming from a commercial online panel. As Table 4 demonstrates, respondents recruited through conversion campaigns were always more likely to pass the attention check, compared to respondents recruited through traffic campaigns. This was most pronounced in Turkey (27% difference) and the Czech Republic (16% difference).

Our second measure of response quality is survey duration. Our initial expectation regarding survey duration, which was also mentioned in our advertisements, was ten minutes. Results were in line with our expectations: the mean value for survey duration was 11.5 minutes while the median value for survey duration was 7.3 minutes. As Table 4 shows, however, there is no clear winner in this regard.

Our third measure of quality is an open-ended question, which asked respondents to write down what they believed "democracy" meant. Respondents had the option of skipping this question. In total, 90% of our respondents wrote at least one word, and the average response

**Table 3. Sample characteristics for conversion and traffic campaigns for survey completers.**

|  | Conversion | Traffic | Population Benchmark |
|---|---|---|---|
| **United Kingdom** | | | |
| *Targeted categories* | | | |
| Young (18–34) | 7%*** | 4%*** | 29% |
| Middle-aged (35–54) | 17%*** | 11%*** | 34% |
| No college | 39%*** | 44%*** | 70% |
| Female | 42%*** | 44%*** | 51% |
| *Non-targeted categories* | | | |
| Less interested in politics | 24%*** | 24%*** | 43% |
| Not partisan | 42%*** | 47% | 48% |
| **Total** | n = 1,033 | n = 756 | |
| **Turkey** | | | |
| *Targeted categories* | | | |
| Young (18–34) | 21%*** | 19%*** | 37% |
| Middle-aged (35–54) | 38% | 27%*** | 40% |
| No college | 32%*** | 73%* | 80% |
| Female | 32%*** | 34%*** | 50% |
| *Non-targeted categories* | | | |
| Less interested in politics | 33%*** | 54% | 48% |
| Not partisan | 42%*** | 56%*** | 32% |
| **Total** | n = 1,689 | n = 203 | |
| **Spain** | | | |
| *Targeted categories* | | | |
| Young (18–34) | 10%*** | 5%*** | 22% |
| Middle-aged (35–54) | 29%*** | 20%*** | 41% |
| No college | 43%*** | 57%*** | 73% |
| Female | 45%*** | 36%** | 52% |
| *Non-targeted categories* | | | |
| Less interested in politics | 17%*** | 28%*** | 60% |
| Not partisan | 22%*** | 34% | 41% |
| **Total** | n = 1,709 | n = 92 | |
| **Czech Republic** | | | |
| *Targeted categories* | | | |
| Young (18–34) | 35%*** | 20% | 25% |
| Middle-aged (35–54) | 40%** | 23%*** | 36% |
| No college | 49%*** | 65%*** | 83% |
| Female | 32%*** | 42%* | 51% |
| *Non-targeted categories* | | | |
| Less interested in politics | 24%*** | 51%*** | 82% |
| Not partisan | 42%*** | 59% | 61% |
| **Total** | n = 1,634 | n = 193 | |

*Note*: Chi-square goodness of fit comparing Meta and population samples:

* = p≤0.05,

** = p≤0.01,

*** = p≤0.001.

The percentages refer to the proportion of respondents as a share of the overall sample in each campaign (column percentages). For example, 42% of respondents recruited through a conversion campaign in the UK are women (and conversely 58% are men). *Population benchmark*: We use data from the round 9 of European Social Survey (for UK, Spain, and CZ) and from the wave 5 of Comparative Study of Electoral Systems (for Turkey) for population benchmarks. We weighted population estimates from these surveys with weight variables provided in the original survey. We used *pspweight* variable for ESS survey and *E1010_2* variable for CSES survey. According to codebooks of these surveys, both of these weight variables were constructed based on demographic information.

**Table 4. Comparison of response quality for campaign objectives for survey completers.**

| | UK | | Turkey | | Spain | | Czech Rep. | |
|---|---|---|---|---|---|---|---|---|
| | Conv. | Traffic | Conv. | Traffic | Conv. | Traffic | Conv. | Traffic |
| Passed attention check | 93% | 92% | 79% | 52% | 84% | 83% | 88% | 72% |
| Median duration (in min.) | 6.6 | 7.4 | 8.5 | 6.4 | 7.5 | 9.6 | 6.4 | 7.3 |
| Mean duration (in min.) | 12.6 | 9.5 | 11.7 | 7.8 | 10.3 | 11.8 | 12.8 | 16.1 |
| Responded open-ended | 95% | 95% | 92% | 65% | 90% | 89% | 87% | 85% |
| Word count in open-ended | 17 | 18 | 12 | 7 | 15 | 15 | 12 | 10 |
| Provided contact details | 68% | 61% | 41% | 22% | 41% | 27% | 51% | 39% |
| Sample size | 1,033 | 756 | 1,689 | 203 | 1,709 | 92 | 1,634 | 193 |

was 14 words long. There are only small differences in responses to this question, except in Turkey.

Finally, at the end of our survey, we asked our respondents to leave their contact information (a WhatsApp number or e-mail address), so that we could re-contact them. Across all the countries studied, respondents recruited through conversion campaigns were more likely to provide contact details than respondents recruited through traffic campaigns. In total, 49% of our respondents provided their phone number or e-mail address. Given that not everyone uses WhatsApp or e-mail and people have privacy concerns, we believe that 49% shows a serious engagement with the survey. For comparison, [32] reports a 62% success rate in collecting email addresses for their research on LGBTQ families in Germany.

To summarize, conversion outperforms traffic and reach campaign objectives in terms of recruitment speed, cost efficiency, and in many of our response quality measures. We do not see major differences in terms of demographic imbalances that could justify continuing to use traffic as a campaign objective. These findings lead us to the conclusion that scholars should use conversion campaign objectives when utilizing Meta advertisements to recruit survey respondents.

## How targeting tools affect participant recruitment

Our analysis in the previous section reveals that significant demographic imbalances might occur in both conversion and traffic samples: older, male, and more educated people are overrepresented in our Meta samples. Targeting tools provided by Meta allow researchers to shape the demographic and political characteristics of their sample. In this section, we discuss best practices to use these tools to produce more representative samples. Since we have already demonstrated a significant cost difference between conversion and traffic campaign objectives, and since traffic campaign objectives failed to create meaningful samples in Turkey, Spain, and the Czech Republic, we limit our analysis in this section to samples recruited through conversion campaigns.

**Overall cost.** Table 5 reveals the trade-off between targeting and survey cost. With the same amount of budget and time, less use of targeting tools usually returns larger samples. The size of the sample created through cross-targeting was only 52% of the sample created through no-targeting in the UK, which means that the average cost per completed survey achieved through cross-targeting is nearly double that achieved with no-targeting. The difference became smaller in other countries, but no-targeting was always cheaper than cross-targeting in terms of average cost of completed surveys. This trade-off nearly vanishes with respect to the comparison between single-targeting and no-targeting, however. In Spain, single-targeting produced cheaper samples than having no-targeting conditions.

**Table 5. Comparison of the cost of conversion-based targeting strategies.**

|  | No-Targeting | Single-Targeting | Cross-Targeting |
|---|---|---|---|
| **United Kingdom (data collection: Feb 9–15, 2021)** | | | |
| Total amount spent | £169 | N/A | £167 |
| Total number of participants recruited | 678 | N/A | 355 |
| Average costs per completed survey | £0.25 | N/A | £0.47 |
| **Turkey (data collection: Feb 19–22, 2021)** | | | |
| Total amount spent | £118 | red£ 118 | £118 |
| Total number of participants | 613 | 597 | 479 |
| Average costs per completed survey | £0.19 | £0.20 | £0.25 |
| **Spain (data collection: Mar 1–15, 2021)** | | | |
| Total amount spent | £499 | £493 | £495 |
| Total number of participants | 584 | 658 | 467 |
| Average costs per completed survey | £0.85 | £0.75 | £1.06 |
| **Czech Republic (data collection: Mar 24-Apr 7, 2021)** | | | |
| Total amount spent | £487 | £487 | £486 |
| Total number of participants | 629 | 553 | 452 |
| Average costs per completed survey | £0.77 | £0.88 | £1.08 |

*Note*: Data collection in each country was stopped when we were about to reach the total of 1,800 respondents across all campaigns in that country.

**Sample representativeness.** In Table 6 we present the proportion of harder-to-reach groups within each of these demographic categories. We conducted chi-square tests to compare the distributional properties of our samples with the population benchmark samples.

A close look at Table 6 reveals how no-targeting strategies can produce cheaper samples: they are significantly less representative than samples using cross-targeting. When demographic targeting is not used, the Meta algorithm is free to channel more resources to demographic groups that are more likely to complete surveys. In our surveys, this group was usually older, male, and college-graduate Meta users.

The demographic imbalances in no-targeting campaigns become even clearer when we look at the intersection of three demographic properties—age, gender, and education, as presented in Section 4 in S1 File. Meta users that are older than 54, male, and college graduates form around 30% of no-targeting samples in the UK, Turkey, and Spain, although this demographic group corresponds to only 2% to 5% of the overall population in these countries. On the contrary, females that are younger than 55 and who do not hold a college degree form less than 1% to 5% of the samples of no-targeting conditions despite forming 20% to 30% of the population in these countries.

These imbalances are corrected to a great extent when we use targeting tools. As Section 4 in S1 File demonstrates, in the samples formed through cross-targeting, all possible combinations of age, sex, and education are represented in the sample to a certain extent. The distributions of these groups within the sample are close to each other and to the distributions within the population, which we calculated based on representative high-quality random samples.

A good convenience sample—as recruited using Meta advertisements—should result in a large sample size with a more representative distribution across demographic groups. Sample representativeness is important not only because we need a certain number of people from each group to produce more generalizable inferences, but also because imbalances in demographic variables can correspond to imbalances in non-observable variables that are hard to control for in a regression analysis. Based on the analysis in this section, we conclude that the

**Table 6. Single group distributions of participants recruited through conversion.**

| | Targeting | | | Population Benchmark |
|---|---|---|---|---|
| | None | Single | Cross | |
| **United Kingdom** | | | | |
| *Targeted categories* | | | | |
| Young | 5%*** | n/a | 11%*** | 29% |
| Middle-aged | 11%*** | n/a | 28%** | 34% |
| No college | 34%*** | n/a | 48%*** | 70% |
| Female | 39%*** | n/a | 49% | 51% |
| *Non-targeted categories* | | | | |
| Less interested in politics | 23%*** | n/a | 27%*** | 43% |
| Not partisan | 43%** | n/a | 41%** | 48% |
| Total | n = 678 | n/a | n = 355 | |
| **Turkey** | | | | |
| *Targeted categories* | | | | |
| Young | 13%*** | 23%*** | 30%** | 37% |
| Middle-aged | 41% | 35%** | 39% | 40% |
| No college | 20%*** | 33%*** | 45%*** | 80% |
| Female | 25%*** | 33%*** | 38%*** | 50% |
| *Non-targeted categories* | | | | |
| Less interested in politics | 31%*** | 31%*** | 37%*** | 48% |
| Not partisan | 40%*** | 42%*** | 44%*** | 32% |
| Total | n = 613 | N = 597 | N = 479 | |
| **Spain** | | | | |
| *Targeted categories* | | | | |
| Young | 4%*** | 10%*** | 17%* | 22% |
| Middle-aged | 23%*** | 26%*** | 40% | 41% |
| No college | 36%*** | 42%*** | 52%*** | 73% |
| Female | 41%*** | 43%*** | 50% | 52% |
| *Non-targeted categories* | | | | |
| Less interested in politics | 17%*** | 15%*** | 21%*** | 60% |
| Not partisan | 23%*** | 21%*** | 21%*** | 41% |
| Total | n = 584 | n = 658 | n = 467 | |
| **Czech Republic** | | | | |
| *Targeted categories* | | | | |
| Young | 38%*** | 36%*** | 29%* | 25% |
| Middle-aged | 42%*** | 38% | 38% | 36% |
| No college | 43%*** | 49%*** | 57%*** | 83% |
| Female | 20%*** | 29%*** | 53% | 51% |
| *Non-targeted categories* | | | | |
| Less interested in politics | 22%*** | 22%*** | 28%*** | 82% |
| Not partisan | 41%*** | 41%*** | 47%*** | 61% |
| Total | n = 629 | n = 553 | n = 452 | |

*Note*: Chi-square goodness of fit comparing Meta and population samples:

* = p≤0.05,

** = p≤0.01,

*** = p≤0.001.

The percentages presented here refer to the proportion of respondents in each of these demographics as a share of the overall sample in each of campaigns (column percentages). For example, 39% of respondents recruited through the non-targeted conversion campaign in the UK are women (and conversely 61% are men).

*Population benchmark*: We use weighted data from the round 9 European Social Survey (for the UK, Spain, and Czech Republic) and from the Comparative Study of Electoral Systems (for Turkey) to create benchmark measures based on representative surveys.

systematic use of targeting tools is necessary to avoid samples greatly overrepresenting older, male, college-graduate Meta users.

**Response quality.**    We do not see any reason to expect a direct impact of targeting strategies on response quality, apart from the indirect effect occurring through the differences in sample compositions. We explore this in more detail in Section 5 of S1 File. As expected, there are only very minor differences in the response quality across samples formed through different targeting strategies.

## Conclusion

In this paper we explore two main tools used to set up advertisements on Meta: campaign objectives and demographic targeting. Our results show that conversion campaigns, which track Meta users outside of the Meta social-media environment, produce the cheapest samples. These samples are no more biased than samples recruited with other campaign objectives, and their data quality is comparable to that of samples recruited through commercial online panels. The only challenge of conversion campaigns is the time needed to set up the very first campaign, which can amount to a couple of hours and requires the ownership of a personal website. For researchers who wish to start data collection immediately or who need only a limited number of participants, traffic campaigns can be an alternative. We have also documented how demographic targeting affects the sample composition. Our results demonstrate that a limited amount of targeting, i.e., single-targeting, can improve sample representativeness without significant financial cost. More aggressive targeting, in which subgroups are crossed with each other, can lead to substantial improvements in sample representativeness despite some increases in cost.

An important limitation of this study, and most other studies on online research tools, is its limited temporal validity [33]. Not only web pages and company policies but also users' online preferences and habits are in constant change. In the case of Meta, the biggest recent change has been Apple's new privacy policy, which allows iOS users to opt out of tracking, limiting Meta's effectiveness at delivering advertisements to iOS users [34]. Users of iOS are still a minority among mobile-phone users; according to StatCounter data, only 29% of mobile phone users around the world have iOS on their devices. This proportion is 55% in the UK, 28% in the Czech Republic, 21% in Spain, and 17% in Turkey [35]. Meta has also introduced a new protocol, called Aggregate Event Measurement, to overcome this challenge. We used this protocol in our research as well. It is not clear how, in the long run, these changes will affect Meta's advertisement system.

Second, Meta regularly makes changes to the way its advertisement system works, and these changes are not always introduced transparently. Most relevant for our study is the recent announcement that campaign objectives will be regrouped throughout 2022 to simplify the advertisement system. Reach campaigns will now be rebranded as awareness campaigns, and conversion campaigns will be rebranded as sales and engagement campaigns. According to Meta, this will not change the way campaign objectives currently work, and advertisers will be able to use the same tools in the ways that are familiar to them [36]. Still, it is important for the community of researchers to follow these changes closely and share their experiences with other researchers using the same tools.

Meta advertisements are a highly valuable research tool for scholars. It is doubtful that commercial online panel companies or crowdsourcing platforms like Amazon Mechanical Turk can provide more diverse or higher quality samples across the world. In Turkey, for example, it took less than four full days for us to reach a sample of 1,800 participants, in which people from all age, education, and gender categories were represented. Benefiting from the

worldwide reach of social media companies can help scholars overcome social science literature's overwhelming focus on WEIRD nations—white, educated, industrialized, rich, and democratic. Participant recruitment through social media also decreases research costs significantly, allowing social scientists with lower research budgets to conduct exciting new research. There are, of course, a series of challenges that are unique to this method of participant recruitment. But we believe that these challenges should only encourage us to conduct more research on the most effective ways to benefit from these powerful tools.

## Supporting information

**S1 File.**
(PDF)

## Acknowledgments

We thank Thomas Gschwend, Francesco Rampazzo, Ericka Rascon Ramirez, participants of the 2021 PolMeth Europe Conference (virtual) and the 2021 General Online Research Conference (virtual), and reviewers and the editor of Plos One for their constructive feedback on this paper.

## Author Contributions

**Conceptualization:** Anja Neundorf.

**Data curation:** Aykut Öztürk.

**Funding acquisition:** Anja Neundorf.

**Methodology:** Anja Neundorf, Aykut Öztürk.

**Project administration:** Anja Neundorf.

**Supervision:** Anja Neundorf.

**Visualization:** Aykut Öztürk.

**Writing – original draft:** Anja Neundorf, Aykut Öztürk.

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
