## [Decision Letter · Decision Letter 0]

14 Feb 2022

PONE-D-21-40688How to Improve Representativeness and Cost-effectiveness in Samples Recruited Through Facebook: A Comparison of Facebook ToolsPLOS ONE

Dear Dr. Neundorf,

Thank you for submitting your manuscript to PLOS ONE. After careful consideration, we feel that it has merit but does not fully meet PLOS ONE’s publication criteria as it currently stands. Therefore, we invite you to submit a revised version of the manuscript that addresses the points raised during the review process.

We look forward to receiving your revised manuscript.

Kind regards,

Yu-Ru Lin

Academic Editor

PLOS ONE

Journal Requirements:

3. We noted in your submission details that a portion of your manuscript may have been presented or published elsewhere. ()

(A preprint of the preliminary results was published here: https://osf.io/3g74n/)

Please clarify whether this publication was peer-reviewed and formally published. If this work was previously peer-reviewed and published, in the cover letter please provide the reason that this work does not constitute dual publication and should be included in the current manuscript.

5. We note that Figure 1 in your submission contain map images which may be copyrighted. All PLOS content is published under the Creative Commons Attribution License (CC BY 4.0), which means that the manuscript, images, and Supporting Information files will be freely available online, and any third party is permitted to access, download, copy, distribute, and use these materials in any way, even commercially, with proper attribution. For these reasons, we cannot publish previously copyrighted maps or satellite images created using proprietary data, such as Google software (Google Maps, Street View, and Earth). For more information, see our copyright guidelines: http://journals.plos.org/plosone/s/licenses-and-copyright.

6. We note that Figure A.1 in your submission contain copyrighted images. All PLOS content is published under the Creative Commons Attribution License (CC BY 4.0), which means that the manuscript, images, and Supporting Information files will be freely available online, and any third party is permitted to access, download, copy, distribute, and use these materials in any way, even commercially, with proper attribution. For more information, see our copyright guidelines: http://journals.plos.org/plosone/s/licenses-and-copyright.

a. You may seek permission from the original copyright holder of Figure A.1 to publish the content specifically under the CC BY 4.0 license. 

Reviewers' comments:

Reviewer's Responses to Questions

**Comments to the Author**

1. Is the manuscript technically sound, and do the data support the conclusions?

Reviewer #1: Yes

Reviewer #2: Yes

2. Has the statistical analysis been performed appropriately and rigorously? 

Reviewer #1: Yes

Reviewer #2: Yes

3. Have the authors made all data underlying the findings in their manuscript fully available?

Reviewer #1: Yes

Reviewer #2: Yes

4. Is the manuscript presented in an intelligible fashion and written in standard English?

Reviewer #1: Yes

Reviewer #2: Yes

5. Review Comments to the Author

Reviewer #1: The manuscript entitled “How to Improve Representativeness and Cost-effectiveness in Samples Recruited Through Facebook: A Comparison of Facebook Tools'' shows the benefits of using various Facebook tools for survey recruitment. The use of Facebook advertisement campaigns has increased dramatically over the past decade, but more research is indeed needed to explore the best practices and benefits of using social media platforms to recruit survey respondents. This work contributes to the literature on online survey methodology by providing concrete and comparative examples of Facebook ads campaigns conducted in four countries utilizing different combinations of features and tools available on the Facebook Ads Manager. Differently from the existing literature and past survey experiments, authors propose to use another feature available on Facebook to reach users who are more likely to complete the questionnaire and therefore compensate for the survey effort and improve the cost-efficiency. Overall, the manuscript is well written, the analysis is rigorous and the literature review is comprehensive and adequate. There are a few minor points I’d recommend clarifying before publication.

- The procedure to link the “Conversation” option to the questionnaire completion (described in the Supplementary Materials) seems much less straightforward than using the “Traffic” option. It seems to require several steps, thus I would recommend the authors to briefly mention it in the main text, focusing especially on the risks of using this option, for example in the context of rapid surveys (e.g. during the first phase of the COVID-19 pandemic) when the data collection needs to be quick and timely. Is there any step in this procedure that may prevent researchers from collecting timely data?

- Structure of the paper: As I said, the manuscript is well written, but it’s quite long and wordy and the structure confusing and sometimes difficult to follow (for example, same headings repeated multiple times). Authors can follow the Plos One guidelines in terms of structure of the manuscript (e.g., Introduction, Material and Methods, Results, etc) to facilitate reading. Given that the results presented here are several, it’s good to have results and discussion together as it already is. However, in the Materials and Methods, I found it often difficult to separate among the literature review, the description of Facebook tools, authors’ speculation and the actual implementation of the study, which is then fundamental to understand the rest of the study. My suggestion is to have two separate sections, one describing the use of Facebook tools to reach survey respondents, including its use in previous studies, and the second one describing clearly and concisely the study design and the methodology tested in the analysis (e.g. combinations of features, targeting, etc).

- Please report the time period of the data collection also in the Methods section when describing the data collection.

- Please report the languages used in the survey experiments and how this may have affected survey responses.

- Page 15: please clarify the variables used in the post-stratification weighting approach

- Please describe the limitations of this study.

- Revise the text for typos, capital letters, and tenses.

Figures and Tables:

- Figure 1: I think it’s ok to show the heterogeneity of Facebook usage across countries and its potential reach. As a clarification, I would suggest separating explicitly those countries with no Facebook users (e.g. Cina) using a different color code in the map (perhaps gray). Not sure whether I missed it in the text, but I would also highlight the proportion of Facebook users in the four countries under study.

- Figure 2 is not very informative, especially since the various options are not described in the text. It can be moved into the Supplementary Material. On the other hand, in my opinion figure A.1 in the Supplementary Material is important when describing the study design. We know from previous studies how different ads materials (e.g. caption, image, etc) may affect survey participation and response, so this represents a rather important element when describing the data collection scheme.

- Table 1 is also not informative, especially because Modes 1 to 8 are not mentioned in the main text. I had to double check again in the text whether I missed their descriptions. It would be much more useful if the table reported the number of participants recruited instead of the labels “Modes”

- Tables 2-6: please double check as symbols % or £ are sometimes missing

Reviewer #2: This article provides guidance to researchers interested in collecting online survey samples via Facebook. The authors evaluate different implementation strategies using surveys collected in four settings -- the UK, Turkey, Spain, and the Czech Republic – with an eye to maximizing representativeness. The key conclusions of the article are that researchers should allow the Facebook algorithms to maximize for the campaign objective of conversion, and that to ensure cost-effective recruitment researchers should target single demographic characteristics of interest (rather than selecting respondents according to multiple demographic features).

In general, I found this to be a thorough and carefully crafted manuscript that provides concrete guidance to a growing body of researchers who are collecting data on the Facebook platform. While the article is highly applied and methods-focused with an eye to a very specific platform, I recommend it for publication because of its potential usefulness to researchers in this field. I suggest minor revisions below among three main lines: (1) discussion and framing; (2) improvements to figures; (3) minor corrections.

Discussion and framing:

• At the superficial level, one could argue that the results of this study are somewhat obvious – if you optimize for conversion, more people will convert, and if you engage in demographic targeting, your sample will be more diverse. Having worked with Facebook advertising myself, I understand why the answers are not obvious and the nuanced analysis in the paper is merited, but I am wondering if you could emphasize a bit the more surprising aspects of your findings (which come out in the main body of the paper, but perhaps could be highlighted more in the abstract, introduction, and conclusion). In my mind, these are:

o Reach is a completely ineffective targeting strategy. On the other hand, conversion targeting does not lead to bias when compared with traffic targeting. In particular, I found the following point really interesting: “it is important to note that traffic samples in these countries are eight to eighteen times smaller than conversion samples, making the absolute number of less educated and female respondents still smaller than conversion samples”.

o That targeting by a single demographic is (cost-wise) competitive with not targeting at all, whereas targeting by multiple demographics is very expensive.

Do you have any sense of why this could be? I am wondering if Facebook might have incomplete demographic data on platform participants, so that by requiring multiple demographic characteristics there might be few people with complete enough records to be considered.

I am wondering if there is a cost-effective strategy here that you could recommend. For example, one could run the survey for a few days with no targeting; then run single targeting for underrepresented categories; then run multiple targeting for categories that are still underrepresented. Would the data you have make it possible to calculate the cost for such a strategy, relative to just doing single or multiple or no targeting?

• At some point in the discussion section I think it needs to be stated that a limitation of the paper is the fact that the Facebook platform is (1) non-transparent and (2) constantly changing. In other words, it is very hard to know how Facebook targets respondents or determines their demographic features, and Facebook’s approaches can change overnight without notice. Therefore there is a possibility that the findings set forth in this paper may not hold true in the future.

• The paper ends pretty abruptly, saying that researchers should experiment with ad images and text. This may be personal preference, but I would like to see a final paragraph that provides more high-level commentary about the themes in the paper and ends on a more inspiring/impactful/meaningful note.

Tables:

• In Table 4, would it be possible to put the number of respondents in the final row? I know this is elsewhere in the paper, but I think having a sense of sample size would help put the differences in perspective.

• For Tables 3 and 6, I found these to be quite dense. As one simple strategy for improving readability, would it be possible to align all of the % symbols (right now the stars push them to the left)? Alternatively, I think it might be preferable to skip a table altogether and illustrate these graphically…

Minor:

• Typos: awarness (p4), poppulation (p11), costed (p14), “and and” (p21)

• Grammatically, I would say “no targeting” instead of “none targeting”

• On p18, when you say “12 versus 7 words”, I would clarify “on average”

• On p21, do you mean “convenience sample” instead of “convenient sample”?

• On p20 (and possibly elsewhere), I would say that “Facebook ADVERTISING is a highly valuable research tool”, rather than “Facebook is”, because I’d think of Facebook as a platform rather than as a tool

• On p20, I would spell out MTurk as “Mechanical Turk” since this is the first time it is mentioned

6. PLOS authors have the option to publish the peer review history of their article (what does this mean?). If published, this will include your full peer review and any attached files.

Reviewer #1: No

Reviewer #2: No

---

## [Author Response · Author response to Decision Letter 0]

24 Jun 2022

Please see the response letter to reviewers.

---

## [Decision Letter · Decision Letter 1]

14 Sep 2022

PONE-D-21-40688R1How to Improve Representativeness and Cost-effectiveness in Samples Recruited Through Meta: A Comparison of Advertisement ToolsPLOS ONE

Dear Dr. Öztürk,

Thank you for submitting your manuscript to PLOS ONE. After careful consideration, we feel that it has merit but does not fully meet PLOS ONE’s publication criteria as it currently stands. Therefore, we invite you to submit a revised version of the manuscript that addresses the points raised during the review process.

We look forward to receiving your revised manuscript.

Kind regards,

Yu-Ru Lin

Academic Editor

PLOS ONE

Journal Requirements:

Additional Editor Comments:

In this revision, the author made substantial changes and I see the manuscript has significant improvement. One reviewer noticed a few presentation issues and suggested the authors carefully check the manuscript.

One of the previous reviewers was not able to re-review the manuscript, so I check the manuscript myself. I found the authors address most of the primary concerns, except that it is unclear whether the question about the language [1] has been properly attended to in this revision.

[1] The review comment: "Please report the languages used in the survey experiments and how this may have affected survey responses."

Based on the reviewer's and my own assessment, I would recommend the authors submit a revised manuscript for consideration.

Reviewers' comments:

Reviewer's Responses to Questions

**Comments to the Author**

1. If the authors have adequately addressed your comments raised in a previous round of review and you feel that this manuscript is now acceptable for publication, you may indicate that here to bypass the “Comments to the Author” section, enter your conflict of interest statement in the “Confidential to Editor” section, and submit your "Accept" recommendation.

Reviewer #2: All comments have been addressed

2. Is the manuscript technically sound, and do the data support the conclusions?

Reviewer #2: Yes

3. Has the statistical analysis been performed appropriately and rigorously? 

Reviewer #2: Yes

4. Have the authors made all data underlying the findings in their manuscript fully available?

Reviewer #2: Yes

5. Is the manuscript presented in an intelligible fashion and written in standard English?

Reviewer #2: Yes

6. Review Comments to the Author

Reviewer #2: The authors have satisfactorily addressed my major concerns and I feel that the paper is clearly written and straightforward. I don't see substantive conceptual barriers to recommending it for publication. However, there are still outstanding issues that should be resolved:

(1) The paper needs a thorough round of copy editing. There are still numerous mistakes and irregularities; I have flagged examples below but there were often more cases in the paper, so I think a careful proofread is necessary:

- "Lead some scholars to questionING" p2.28

- Subject-verb disagreement: "data collection through Meta platforms HAS" p2.42, "reach campaigns emergE" p3.63, "Table 4 demonstrateS," p10.371

- Unnecessary hyphens: e.g. "data-collection methods", "survey-completers", "link-clickers", "set-up"

- The word after a colon should not be capitalized

- Missing articles: e.g. "outside of THE Meta environment" p3.55, "THE Meta algorithm" p3.91, "on THE Qualtrics survey platform" p4.98, "all of THE targeted demographic categories" p4.118, "finally, THE Czech Republic" p5.147, "THE purposes of our research p6.199, "shown to THE most Meta users....provide THE most link clicks" p8.277, "seen by THE most Meta users" p8.281, "that THE reach campaign objective" p8.287

- I believe p4.105 should read "with reference to convention"

- I believe p4.115 and p4.141 should read "demographic characteristics", not "properties"

- I believe p5.175 should read "old", not "old-aged"

- I believe p6.214 "the divide of..." should be rephrased

- I believe p10.329 should read "AMOUNT of traffic"

(2) On p3.66, the authors state that "the former is usually not even more expensive than the latter" - I would specify "in our experience", because it is hard to conclude this more generally.

(3) Meta is misquoted on p3.83; this does not match the text in the source.

(4) On p4.110, "you can" sounds a bit too casual to me.

(5) The text refers to Turkey as a "non-democratic country"; while it is not an open society, I believe that factually (technically) it is considered a democracy. Similarly, the paper implies that Spain is between the UK and Turkey in terms of political openness, but is this true? In my understanding it would be as open/democratic as the UK.

(6) On p5.162 you say that "campaigns from the same country stayed open for the same amount of time", but Figure 2 suggests otherwise.

(7) I think Table 2 would be clearer if the left column (A/B/C) were dropped and the italicized entries were called "click-through rate" and "completion rate". Then, the note at the bottom of the table could explain how these were calculated.

(8) On page 18, the description of the weights is missing some punctuation/explanatory words.

(9) On p9.378, the table says that participants recruited through conversion spent more time on average, but this is only true in 2/4 contexts.

Based on the number of errors identified, I would suggest that the paper be carefully checked again before it is approved for publication.

7. PLOS authors have the option to publish the peer review history of their article (what does this mean?). If published, this will include your full peer review and any attached files.

Reviewer #2: No

---

## [Author Response · Author response to Decision Letter 1]

15 Oct 2022

We are grateful that both the editor and the Reviewer 2 went through our manuscript carefully and pointed out some sufficient language in our manuscript. We corrected these mistakes, and the manuscript is in a better form now. Furthermore, we have also had the entire manuscript proofread by a professional proofreader. Below we list all the changes we made during the second revision process. 

Editor comments: 

1: Editor: "Please report the languages used in the survey experiments and how this may have affected survey responses."

We have conducted all of the surveys in official and the most spoken languages in those countries. This has increased the size of our audience in each country. This practice is also the norm in the political science discipline. 

In response to this comment, we have added a new sentence on page 4: “All of the surveys were conducted in the official and the most spoken language of the respective country, i.e., English, Turkish, Spanish, and Czech, allowing us to maximize the size of our audience.”

Reviewer 2’s comments: 

1: First, Reviewer 2 points out a series of language mistakes, which have now all been corrected. 

2: “On p3.66, the authors state that "the former is usually not even more expensive than the latter" - I would specify "in our experience", because it is hard to conclude this more generally.”

We have added “in our experience” to the referred sentence. 

3: “Meta is misquoted on p3.83; this does not match the text in the source.”

Thank you for noting this problem. We have realized that Meta had made a change to the URL link. We have updated the URL link. 

4: “On p4.110, "you can" sounds a bit too casual to me.”

Instead of “you can see,” we have inserted “please see.”

5: “The text refers to Turkey as a "non-democratic country"; while it is not an open society, I believe that factually (technically) it is considered a democracy. Similarly, the paper implies that Spain is between the UK and Turkey in terms of political openness, but is this true? In my understanding it would be as open/democratic as the UK.”

According to V-Dem and Freedom House classifications, which are used by most of the political scientists, Turkey is currently ruled by an electoral authoritarian regime. On the other hand, the reviewer is right that today’s Spain is as democratic as the UK, although democratic institutions are younger in this country. We have revised the relevant paragraph to focus on the level of economic development in Spain and the Czech Republic and add a reference to Varieties of Democracy dataset:

“The UK constitutes a typical case of an advanced democracy, and it is the second most-studied country in political science according to Wilson (2020). However, many researchers are increasingly using Meta advertisements to study less developed, non-democratic countries, where data collection is more restricted. Turkey represents such a context. It is a middle-income country that was, according to the influential Varieties of Democracy dataset, an electoral authoritarian regime by the time our study was conducted. Spain and the Czech Republic are younger democracies that are located between the UK and Turkey in respect of their levels of gross domestic product (GDP) per capita. The Czech Republic is also distinguished by its relatively small population, which amounts to only 10 million people.”

6: “On p5.162 you say that "campaigns from the same country stayed open for the same amount of time", but Figure 2 suggests otherwise.”

We believe that we were not clear enough in our sentence. What we meant is that each of the campaigns run in a country stayed open for the same amount of time. Thus, for example, all campaigns in the UK stayed open for 5 days and all campaigns in Turkey stayed open for 4 days. This is what Figure 2 shows as well. 

To make this clearer, we revised the sentence. It now reads: “each of the campaigns running in a country stayed open for the same amount of time.”

7: “I think Table 2 would be clearer if the left column (A/B/C) were dropped and the italicized entries were called "click-through rate" and "completion rate". Then, the note at the bottom of the table could explain how these were calculated.”

We made the changes in Table 2 as suggested by Reviewer 2. 

8: “On page 18, the description of the weights is missing some punctuation/explanatory words.”

Page 18 included the bibliography in the previous manuscript. Yet, we used to refer to weights in page 11, saying: “We weighted these benchmarks with weights included in these surveys. We used pspweight for ESS E1010 2 for CSES. Both of these weights are constructed based on demographic information.” We believe this is what the reviewer means. 

We have updated this sentence to make it clearer. The new version reads: “We weighted population estimates from these surveys with weight variables provided in the original survey. We used pspweight variable for ESS survey and E1010_2 variable for CSES survey. According to codebooks of these surveys, both of these weight variables were constructed based on demographic information.”

9: “On p9.378, the table says that participants recruited through conversion spent more time on average, but this is only true in 2/4 contexts.”

Reviewer 2 is right in pointing out that in half of the countries participants recruited through traffic spent more time on average. We revised the sentence accordingly. The new sentence reads: “As Table 4 shows, however, there is no clear winner in this regard.”

---

## [Editor Report · Decision Letter 2]

19 Jan 2023

How to Improve Representativeness and Cost-effectiveness in Samples Recruited through Meta: A Comparison of Advertisement Tools

PONE-D-21-40688R2

Dear Dr. Öztürk,

We’re pleased to inform you that your manuscript has been judged scientifically suitable for publication and will be formally accepted for publication once it meets all outstanding technical requirements.

Kind regards,

Yu-Ru Lin

Academic Editor

PLOS ONE

Additional Editor Comments (optional):

I have checked the authors' response and revision myself, to avoid further delay on the reviewers' end. I think the authors have satisfactorily addressed the review comments from the previous round, and I'll recommend it for publication. 

I do have a minor suggestion for the authors. In response #6, the current text reads: "each of the campaigns running in a country stayed open for the same amount of time." I think the current text still seems to be confusing. To bring sufficient clarity, I'd suggest something like: "all campaigns running in the same countries stayed open for the same amount of time, but the open durations of campaigns running in different countries may vary -- e.g.,  all campaigns in the UK stayed open for 5 days, and all campaigns in Turkey stayed open for 4 days."

---

## [Editor Report · Acceptance letter]

30 Jan 2023

PONE-D-21-40688R2 

How to Improve Representativeness and Cost-effectiveness in Samples Recruited through Meta: A Comparison of Advertisement Tools 

Dear Dr. Öztürk:

I'm pleased to inform you that your manuscript has been deemed suitable for publication in PLOS ONE. Congratulations! Your manuscript is now with our production department. 

Kind regards, 

on behalf of

Dr. Yu-Ru Lin 

Academic Editor

PLOS ONE